# Improving Patients’ Life Quality after Radiotherapy Treatment by Predicting Late Toxicities

**DOI:** 10.3390/cancers14092097

**Published:** 2022-04-22

**Authors:** Ariane Lapierre, Laura Bourillon, Marion Larroque, Tiphany Gouveia, Céline Bourgier, Mahmut Ozsahin, André Pèlegrin, David Azria, Muriel Brengues

**Affiliations:** 1IRCM, INSERM, University Montpellier, ICM, 34298 Montpellier, France; ariane.lapierre@chu-lyon.fr (A.L.); laura.bourillon@icm.unicancer.fr (L.B.); marion.larroque@icm.unicancer.fr (M.L.); tiphany.gouveia@inserm.fr (T.G.); celine.bourgier@icm.unicancer.fr (C.B.); andre.pelegrin@inserm.fr (A.P.); david.azria@icm.unicancer.fr (D.A.); 2Department of Radiotherapy-Oncology, Lyon-Sud Hospital Center, 69310 Pierre-Bénite, France; 3CHU Vaudois, 1011 Lausanne, Switzerland; mahmut.ozsahin@chuv.ch

**Keywords:** biomarkers, radiotherapy, late toxicities prediction, personalized treatment

## Abstract

**Simple Summary:**

Over 50% of patients with cancer will receive radiotherapy treatment. Five to ten percent of patients who received radiotherapy will develop side effects. Identifying these patients before treatment start would allow for treatment modification to minimize these effects and improve the life quality of these patients. Our team developed a test, which allows predicting these secondary effects before starting the treatment. This will help in proposing personalized treatments to improve the outcome. This review presents how this test is performed, its results, as well as its modification in order to be used in hospitals.

**Abstract:**

Personalized treatment and precision medicine have become the new standard of care in oncology and radiotherapy. Because treatment outcomes have considerably improved over the last few years, permanent side-effects are becoming an increasingly significant issue for cancer survivors. Five to ten percent of patients will develop severe late toxicity after radiotherapy. Identifying these patients before treatment start would allow for treatment adaptation to minimize definitive side effects that could impair their long-term quality of life. Over the last decades, several tests and biomarkers have been developed to identify these patients. However, out of these, only the Radiation-Induced Lymphocyte Apoptosis (RILA) assay has been prospectively validated in multi-center cohorts. This test, based on a simple blood draught, has been shown to be correlated with late radiation-induced toxicity in breast, prostate, cervical and head and neck cancer. It could therefore greatly improve decision making in precision radiation oncology. This literature review summarizes the development and bases of this assay, as well as its clinical results and compares its results to the other available assays.

## 1. Introduction

Radiotherapy is one of the leading treatment modalities in oncology. Over 50% of patients will receive radiotherapy at some point during their treatment course [1]. Although it is a locoregional treatment, patients can exhibit toxicities in the treatment field or in the surrounding tissues. These toxicities can be defined either as early (occurring during or in the 3 months after treatment completion) or late (occurring more than 3 months after treatment completion). Depending on the prognosis and tumor type, the prescription dose and constraints to organs-at-risk are usually chosen in order to keep the risk of developing grade 3 or higher side effects below 5% [2,3]. However, even when keeping these constraints, 5 to 10% of patients will develop sever toxicities after radiotherapy. 

In breast cancer, severe toxicities can present as breast or lung fibrosis. In cerebral radiation therapy, cerebral radiation necrosis is the most frequent occurrence. In pelvic and abdominal radiotherapy, severe toxicities can be radiation enteritis and vesical or rectal bleeding. 

Patients displaying severe toxicities can be considered intrinsically radiosensitive [4]. The first clinical observation of individual radiosensitivity was described by Holthusen in 1936 [5], whereas the first in vitro display of individual radiosensitivity was shown on fibroblasts of ataxia telangiectasia patients in 1975 [6].

Early toxicities can usually be managed using symptomatic treatments and will most of the time resolve after treatment completion. On the other hand, late toxicities can be definitive, and severely affect quality of life, sometimes requiring extensive treatments such as surgery to alleviate the symptoms. Based on these observations, it appears crucial to identify the patients at risk of developing severe late toxicities early on, because severe toxicities in a minority of patients limit the dose for the majority of patients [7]. Furthermore, these patients need to be identified before treatment starts, because acute toxicities may not always predict late toxicities [8].

The first large scale clinical search of individual factors of radiosensitivity was performed in the 1970s by Turesson et al. [9]. However, in this study, clinical factors and early toxicities only explained 30% of late toxicities, leaving 70% unexplained. Although influenced by many exogenous factors (such as smoking habits, age or ongoing treatments), it seems rather unlikely that individual radiosensitivity should be caused by only one intrinsic factor. It seems reasonable to assume that clinical radiosensitivity should be regarded as a complex trait depending on the combined effect of several different genetic alterations [10]. Should these genetic traits be successfully found, early identification of patients at risk of severe late toxicities could allow physicians to suggest a more appropriate treatment course (such as radical mastectomy instead of conservative breast surgery) in cases where the risk of toxicity outweighs the benefits of the radiation treatment [11,12]. In the near future, this could lead to tailored treatment based on the risk profile of each patient, adapting treatment dose or technique to each individual situation. More recently, in the 2000s, several genetic profile studies have come up with gene expression models linked to tumor radiosensitivity in vitro [13,14]. When looking at healthy tissue toxicities, genomic signatures, single nucleotide polymorphisms (SNPs) variability, or apoptosis or cell cycle regulating gene expression changes after irradiation appear to have better potential at classifying patients [15,16]. Even though it has been widely discussed for over 20 years [17], the American Society for Radiation Oncology (ASTRO), the American Association of Physicists in Medicine (AAPM) and the National Cancer Institute (NCI) have recently established guidelines on precision medicine in radiotherapy, mainly for breast, prostate, lungs and head and neck cancers [15]. Their conclusion is that genomically guided radiation therapy is a necessity that must be embraced in the coming years, to improve outcomes for numerous cancer patients. However, routine genomic signature and clinical tests still need to be brought into routine standard of care.

## 2. Development of the Radiation-Induced Lymphocyte Apoptosis (RILA) Assay

The first correlation between in vitro assays and clinical findings was performed using skin fibroblasts of ataxia telangiectasia patients [6]. In this study, the authors observed a difference in in vitro response to radiotherapy of primary fibroblast cultures between ATM-mutated patients and healthy controls, showing that in vitro observation could translate to the clinic. Further studies, based on colony-forming assays or surviving fraction at 2 Gray (Gy) (SF2) showed a strong relation between fibroblast sensitivity in vitro and normal-tissue reactions, both acute effects and late fibrosis [18,19]. However, although these results were promising, both studies were performed on small groups of patients (respectively, 6 and 12 patients), and further validation on larger cohorts was needed to confirm these observations. Unfortunately, when performed on a larger group (79 patients), no significant correlation between fibroblast radiosensitivity and fibrosis could be found, because of significant inter-patient variation for SF2 values (over 40%) [20]. Other fibroblast-based assays such as comet assays or micronuclei formation were investigated [21,22]. However, in both cases, despite promising results in small study groups, no significant correlation was found between these in vitro tests and patient radiosensitivity in larger cohorts [23]. Based on these observations, and given the fact that fibroblast radiosensitivity assays have a long completion time (over one month), a simpler and more reliable in vitro assay was needed. Since fibroblasts assays were rather time-consuming, researchers turned to easily available cells: peripheral blood mononucleated cells (PBMCs). Out of these PBMCs, lymphocytes were soon selected as a study model because of their higher radiosensitivity compared to other cell types [24].

The first studies investigating peripheral lymphocytes irradiation gave inconsistent results [25,26,27]. Although comparing cell survival after irradiation, irradiation was performed at low-dose rate in all three studies. Lacking a clear standard for their tests, inter-patient and intra-patient variations were very high and no correlation to the clinic could be found because of the lack of reproducibility. High dose-rate irradiation for in vitro studies started to develop in the 1990s. At first, the assays used were the same as the fibroblast-based assays: colony formation, SF2, comet and micronuclei assays. Once again using ataxia telangiectasia patients, West et al. showed that peripheral blood lymphocytes from patients who suffered severe reactions to radiotherapy were more radiosensitive than those from normal donors [28]. However, micronuclei assay data showed large discrepancies between studies and no clear conclusion could be made [29,30,31]. The same goes for comet assays: although the test could identify patients with defective in vitro DNA repair mechanisms, no correlation could be made between these findings and radiation-induced toxicities in patients [32]. However, analysis of lymphocyte apoptosis after irradiation showed a different response to radiotherapy in patients with genetic disorders such as ataxia telangiectasia of neurofibromatosis when compared to healthy counterparts [33]. Apoptosis may not be the predominant death type after radiotherapy in most cancer cell lines; however, it is much more frequent in hematopoietic cell lines such as peripheral lymphocytes [34]. This particular cell death mechanism occurs rapidly after irradiation (6 to 72 h) and can be easily detected by flow cytometry [35]. Therefore, in the 1990s, Ozsahin et al. developed a rapid assay to detect peripheral lymphocyte apoptosis after irradiation [36]. This assay was based on the analysis of apoptosis of both CD4 and CD8 T-lymphocytes 48 h after 8 Gy irradiation using flow cytometry. The result was given as a percentage of apoptosis at 8 Gy, subtracting the apoptosis at 0 Gy (non-irradiated samples) as a control (Figure 1). CD4 and CD8 T-lymphocytes apoptosis was correlated in all adult donors, and inter-donor variations were higher than intra-donor variations, displaying a good reproducibility of this assay. This was later named the radiation-induced lymphocyte apoptosis (RILA) assay.

Blood samples were collected from donors in Heparin tubes, diluted in RPMI medium (1:10) and then cultured in 6-wells plate at 37 °C, 5% CO_2_ for 24 h prior to ex-vivo irradiation (0 or 8 Gy). Irradiated whole blood was cultured for 48 h, red blood cells were lysed and the remaining cells were labeled with FITC-conjugated anti-CD8 monoclonal antibodies to select CD8 + T-lymphocytes that were then stained with propidium iodide (PI). Cells were analyzed by flow cytometry to determine the percentage of apoptotic cells.

## 3. Clinical Data

The first prospective study using this RILA assay followed 399 patients with miscellaneous cancers (mostly breast, head and neck, genitourinary and gastrointestinal) treated with radiotherapy with curative intent [37]. The CD4 and CD8 RILA assays were performed before radiotherapy, and patients were assessed for both acute and late toxicity. With a median follow-up of 30 months, T-lymphocyte radiation-induced apoptosis did not correlate with either early toxicity or survival. However, more radiation-induced T-lymphocyte apoptosis was significantly associated with less grade 2 and 3 late toxicity (*p* < 0.0001). CD8-RILA was more sensitive and specific than CD4-RILA, and thus from this point on, most studies used CD8 T-lymphocytes apoptosis for the RILA assay. 

This was confirmed in a phase II multicenter prospective study: the CO-HO-RT trial [38]. A total of 150 breast cancer patients were tested with the RILA assay before breast adjuvant radiotherapy. With a median follow-up of 26 months, high RILA scores (i.e., a high level of CD8-T-lymphocyte apoptosis after 8 Gy irradiation) proved once again to be associated with fewer grade 2 or more toxicities. A longer follow-up of these patients, as well as another prospective multicenter study on 502 breast cancer patients, confirmed these results [39,40]. In both studies, a RILA score over 12% was significantly associated with lower grade 2 or more late breast fibrosis (*p* = 0.012). However, in these studies, late fibrosis was also correlated with hormonotherapy and, although both hormonotherapy and RILA independently influenced late breast fibrosis, RILA appeared to be a continuous risk-variable rather than a high or low risk discrete variable [41]. A recent review of the significance of the RILA in breast cancer summarizes these results [42].

RILA has also been assessed in two small prospective studies in cervical and head and neck cancer [43,44]. In both cases, a high RILA score was associated with lower severe late toxicities. Larger studies have been published on prostate cancer, using both CD4 and CD8 T-lymphocytes [45,46,47]. In all three studies, a higher RILA score was significantly associated with a lower-risk late toxicity. However, with rather small patient samples (45, 12 and 50 patients, respectively), the results were inconsistent between studies, one showing significantly lower genito-urinary toxicity, where the other only showed lower gastro-intestinal toxicities [46,47]. However, in a more recent prospective multicenter trial on a larger population (383 patients), a RILA score over 15% was associated with lower grade 2 or more toxicities, both genito-urinary and gastro-intestinal, confirming both earlier studies’ results [48]. Other cancer types, such as lung cancer, are currently being tested as part of a wide multicenter trial: the REQUITE project [49,50,51]. This study, including breast and prostate cancer patients, should also further validate the data already published on these cancer types.

A summary of published studies and results by cancer types can be found in Table 1.

All of these data suggest that a high RILA score is associated with a low risk of late toxicity after radiotherapy. Subsequently, low-RILA patients should be considered at higher risk of developing severe late toxicity after radiotherapy, and alternate treatment should be considered when available. For example, mastectomy could be proposed to patients with localized breast cancer in order to forgo postoperative radiotherapy. In cases where radiotherapy is still warranted but the patient has a high risk of severe toxicity, fractionation could be altered to protect healthy tissues. On the other hand, in the case of high-risk tumors in patients with a low risk of severe toxicities, treatment could be escalated by adding concurrent chemotherapy. Other treatment alterations are suggested in Azria et al. [12]. However, since no strong correlation has been found between low RILA and an increased risk of radiation-induced toxicities, radiotherapy should be maintained when it is the standard of care. Although the mechanism of this inverse association is not completely clear, it may possibly be related to the delay of cells in recognizing the radiation-induced cell damage and initiating apoptosis, with a consequently increased risk of toxicity and, theoretically, of cancer radioresistance and reduced tumor control for low-RILA patients [52]. However, to date, no correlation between low RILA values and low tumor control has been described in the literature. The RILA assay has been used in numerous studies, in various centers and countries. Where earlier radiosensitivity assays had low reproducibility, this test is robust, and its results have been confirmed in different centers with similar results for same patients, further validating its use in different centers [53,54].

Although prospective data to predict toxicities were similar between all studies, one retrospective study found rather contradicting results in prostate cancer [55]. This was a retrospective analysis of the Epinal radiation incident, where 409 prostate cancer patients received over 108% of the prescribed dose due to overexposure related to portal imaging. In this analysis, RILA did not correlate with inter-individual variations in maximum digestive or urinary toxicity. However, in this case, the magnitude of the overdosage may override the biological predictors of toxicity, including individual radiosensitivity. More interestingly, a prospective study investigating 120 patients who developed radiation-induced sarcomas (RIS) found that patients with a high RILA value were less likely to develop RIS. In this matched cohort study, the mean RILA value was lower in RIS than in control patients (18.5% vs. 22.3%, *p* = 0.0008), and patients with a RILA > 21.3% were less likely to develop RIS (*p* < 0.0001) [56].

In summary, with prospective data available in different clinical settings, the RILA assay shows great promise in predicting long-term toxicities after radiotherapy.

## 4. Molecular Rationale for the RILA Assay

The molecular bases underlying the RILA assay are still unclear. Even though the mechanisms leading to radiation-induced fibrosis have been thoroughly investigated [57], the role of peripheral lymphocytes, specifically CD8 T lymphocytes remains unknown. However, some new hypotheses are starting to rise in an attempt to explain the differences of radiation-induced lymphocyte apoptosis among patients. 

Apoptosis does not appear to be the most important mode of cell killing by radiation in most cases in vitro and in vivo but it has been described in both tumor cells and normal tissues after irradiation. Although mitotic death is usually described as being the preferential mode of radiation-induced cell death in proliferating cells, several studies have shown that apoptosis may be induced preferentially in the S phase of the cell cycle [58]. However, in mature lymphoid cells and lymphocytes, apoptosis appears to be the leading cell death mechanism after irradiation. The role of apoptosis in normal tissue response to radiation has been investigated using p53-deficient mice. In this model, there is an increased survival of haemopoietic cells and fibroblastoid stromal precursor cells after irradiation, due to a larger shoulder in the survival curves [59]. This shows that a decrease in apoptosis affects not only apoptotic prone cells, but other tissues as well. Furthermore, the wider shoulder in survival curve could be correlated to increased DNA repair, but this may lead to increased acquired mutations and alter cell function overtime. As such, patients displaying lower levels of radiation-induced apoptosis in their lymphocytes may exhibit greater radioresistance in their connective tissues as well, leading to delayed reaction to radiation, such as fibroblast proliferation leading to fibrosis.

CD8 T lymphocytes have been shown to produce basic Fibroblast Growth Factor (bFGF), while CD4 T lymphocytes produce both bFGF and heparin-binding epidermal growth factor-like growth factor (HB-EGF) [60]. These cytokines are potent mitogens for fibroblasts and endothelial cells. Furthermore, bFGF has been shown to protect endothelial cells from radiation-induced cell death both in vitro and in vivo [61]. As such, patients with decreased T cell apoptosis after radiation may have increased production of fibroblast growth factors, increasing radiation resistance and proliferation of fibroblasts in the treated region.

Another hypothesis is that patients for whom a severe and late radio-induced side effect is associated with a low RILA, may have a pool of lymphocytes more resistant to radiation-induced apoptosis, which could therefore reflect the presence of cells in senescence that will be participating in the development of these late effects in the irradiated healthy tissue (fibroblasts). Ionizing radiations can induce a variety of cell death responses including apoptosis, but also senescence. Senescent cells have reduced sensitivity to apoptosis, and a pro-inflammatory secretory phenotype. In addition, ionizing radiations can induce the production of reactive oxygen species (ROS) that cause DNA damage in non-targeted tissue, and systemic effects associated with inflammation. 

It has recently been shown that, in healthy donors Th17 CD4 T lymphocytes are less sensitive to apoptosis and more sensitive to senescence than other subtypes of CD4 T lymphocytes [62]. Other groups have observed a high frequency of Th17 cells in murine radiation induced pneumonitis with fibrosis, in comparison with pneumonitis without fibrosis [63]. It has also been shown that the balance between Th17 and regulatory T lymphocytes (Treg) might modulate radiation induces lung fibrosis [64]. It can thus be hypothesized that patients with a low RILA value might have an imbalance in their Th17 ratio.

In conclusion, the molecular rationale for the RILA assay is still very much unclear, but several hypotheses point towards a correlation between peripheral lymphocytes and radiation induced fibrosis. A summary of the hypotheses can be found in Figure 2.

## 5. RILA Compared to Other Radiosensitivity Assays

As discussed above, RILA tests have been performed on different cell populations. Where the first CD4 results were less reproducible than CD8 results, a recent study on 272 breast cancer patients with over 10 years of follow-up showed that low CD4-RILA was associated with increased risk for both fibrosis and telangiectasia [65]. However, in this study, neither CD8 nor NK-RILA were correlated with late toxicity. Another comparison between CD4, CD8 and NK-RILA in breast cancer patients showed that both CD8 and NK lymphocytes were associated with late toxicity [66]. A last study compared CD8 RILA to CD4 and B-lymphocyte RILA in 94 cervical cancer patients [67]. In this study, both CD8 and B-lymphocyte RILA were significantly correlated with toxicities, whereas CD4-RILA was not.

Overall, RILA seems to be applicable to different lymphocyte populations. However, as the largest studies were published using CD8-T-lymphocytes, the standard cell population for this assay remains CD8 lymphocytes.

As seen before, RILA seems a robust and reproducible test to assess the risk of late radiation-induced toxicities and delayed complications in various cancer types. However, it seems important to compare it with other available radiosensitivity assays. As the only assay tested in a prospective multicenter study, RILA qualifies as the highest level of evidence. Only the SNP analysis in prostate cancer can also be considered level I, since a large meta-analysis has confirmed the link between their expression and radiosensitivity [68].

A summary of the different assays and their level of evidence in shown in Table 2.

In breast cancer patients, RILA was compared to other lymphocyte-based assays: residual DNA double-strand breaks (DSB), G0 and G2 micronucleus assay [70]. In this case-control study, the RILA assay performed best to detect individual radiosensitivity, with a strong correlation between the RILA value and the clinical outcome (*p* < 0.01), followed by the residual DSB and both micronuclei assays. The same results were shown in prostate cancer patients. When comparing RILA to γ-H2AX and G2 micronuclei assays, lymphocyte apoptosis analysis appeared to be the most suitable test for patients’ radiosensitivity prediction [46]. In breast and head and neck cancer patients, CD3-lymphocyte radiation-induced apoptosis was compared to DNA strand breaks (Comet assay), γ-H2AX foci, and whole genome expression analyses [88]. Once again, inter-individual variations and inter-laboratories variation were very high for most of these tests, although lymphocyte apoptosis seemed the most robust assay. Initial DNA damage, measured by DSB, was also compared to RILA data in 26 breast cancer patients [90,91]. In this study, patients who presented lower levels of initial DNA damage had higher RILA scores and fewer late toxicities. The two assays’ results seemed correlated; although, the patient sample was small and a prospective analysis is still required to confirm those results. The only other radiosensitivity assay with a high level of evidence is the SNPs analysis for prostate cancer [68,92,93]. In 2008, Azria et al. compared RILA results and these known SNPs variability in late radiation-induced toxicity prediction in 399 patients with miscellaneous cancers [94]. In the low-RILA (<9%) patient group, where patients had higher toxicity rates, 94% of patients had four or more SNPs, whereas in the high-RILA group, only 33% had four or more SNPs. Although the numbers are rather small in this study, this points towards a good correlation between the two most robust assays for assessing individual radiosensitivity. Overall, with a higher level of clinical evidence than most tests, the RILA assay appears to be one of the most robust tests and its results correlate to other available radiosensitivity assays. Furthermore, cost wise, the RILA is a relatively cheap assay, around EUR 150 per test, making it easy to implement in a clinical routine; although, most available tests have a similar price range. 

Overall, with a higher level of clinical evidence than most tests, the RILA assay appears to be one of the most robust tests and its results correlate to other available radiosensitivity assays. 

## 6. Use of RILA in Clinical Routine

Due to considerable progress in cancer management in recent decades, the number of cancer survivors has dramatically increased, raising new challenges in the various phases of survivorship. Thus, post-treatment morbidity and quality of life have become a critical concern in the growing patient population. The medico-economic consequences of severe late side effects can also be consequential, as treatments to alleviate the symptoms range from lifelong pain medication to major surgery. The ultimate goal of any radiosensitivity assay is thus to identify the patients at risk for severe toxicity before radiation treatment to offer therapeutic alternatives. These depend on two main factors: tumor control probability (TCP) and normal tissue complication probability (NTCP). In the case of low-risk tumors, patients at risk for severe toxicity could be offered surveillance instead of radiation, or smaller fields of radiation. However, when tumor control is critical, alternative treatments such as surgery or chemotherapy should be discussed. A list of possible treatment adaptations based on TCP and NTCP has been proposed by Azria et al. [12]. 

Although alternatives to radiation therapy exist in many cases, when radiation is the standard of care, the radiation course can be tailored to fit the patient’s individual radiosensitivity. Clinical trials studying fractionation schedule alteration or long-term toxicities prevention through additional drugs are currently ongoing (NCT04282122, NCT04385433). Another aspect currently under investigation is the cost-utility of these models. This is being carried out in Europe through the ongoing REQUITE project, using the RILA assay, as well as other validated biomarkers [51].

In summary, although radically changing a treatment course based simply on radio-sensitivity assays should not be undertaken outside of clinical trial settings, available alternatives should be proposed when available and validated.

## 7. Conclusions

Identifying patients at risk of severe radiation-induced toxicity before treatment is one of the cornerstones of precision medicine applied to radiotherapy. Although numerous assays have been developed over the last few decades, only a couple reach the highest level of evidence: SNPs analysis in prostate cancer and the RILA assay in several cancer types. Out of these, the RILA assay seems the easiest to use in clinical routine, especially without the need of using an X-ray irradiator like in the original version of RILA. By replacing the irradiation step by the addition of bleomycin, the procedure becomes transferrable in any clinical laboratory. The procedure itself is rather simple and results can be obtained under a week’s time. To date, the RILA has been validated in breast, prostate, cervix, head and neck cancer, and validation in lung cancer is pending. 

Although the mechanistic basis of this test still remains unclear, the RILA assay appears to be a robust help in deciding the best treatment course in radiotherapy planning. Taking into account tumor prognosis as well as late results and quality of life, the RILA assay, incorporated in a nomogram with the other independent factors, can be used safely in a clinical setting. Wider use of this test would allow for a personalized risk-adapted approach to provide more effective treatments for patients receiving radiotherapy.

In case of high local relapse risk and low toxicities risk (high RILA value), new strategies could be considered as an increase in the total dose; in case of high local relapse risk and high toxicities risk (low RILA value), indication of radiotherapy should be discussed and alternative locoregional treatments should be preferred; in case of low local relapse risk and high toxicities risk (low RILA value), antifibrotic agents could be recommended in a mitigation approach in order to prevent or reduce the severity of late radio-induced toxicities (PRAVAPREV study, ClinicalTrials.gov Identifier: NCT04385433).

## 8. Future Directions

Despite the clinical evidence, mechanistic rationale of the RILA assay remains uncertain. Further research is still warranted to identify the role of lymphocyte apoptosis in the development of fibrosis after radiation treatment.

From a clinical point of view, the cost-utility of such markers is still under study, and the ongoing REQUITE project should shed a light on this aspect in the next few years. Their relevancy in clinical routine is also being assessed through two clinical trials studying adapted treatment modalities (fractionation schedule alteration or long-term toxicities prevention through additional drugs): NCT04282122, NCT04385433.

## Figures and Tables

**Figure 1 cancers-14-02097-f001:**
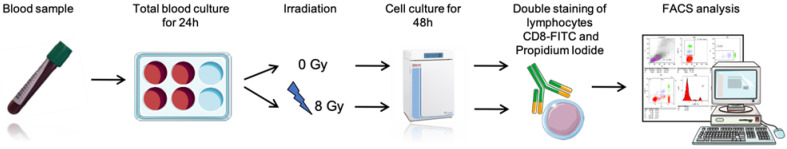
RILA assay procedure (adapted from Brengues et al. [36]).

**Figure 2 cancers-14-02097-f002:**
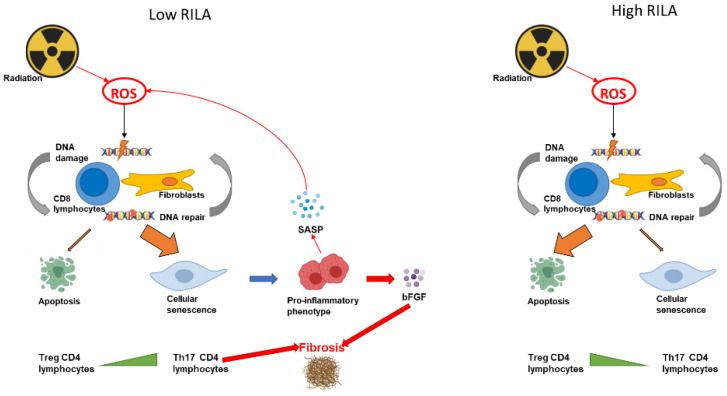
Main hypotheses for RILA molecular rationale.

**Table 1 cancers-14-02097-t001:** Available data on RILA assay by tumor type. GU: genito-urinary, GI: gastro-intestinal.

Tumor Type	Data Type	Patient Number	Results	References
Breast	Prospective multicenter	577	Correlation with fibrosis (RILA cutoff = 12%) (*p* = 0.001)	[39,40,41]
Prostate	Prospective multicenter	692	Correlation with GU and GI toxicity (RILA cutoff = 15%) (*p* = 0.01)	[45,46,47,48]
Cervix	Prospective	94	Correlation with sexual toxicity (*p* = 0.001)	[43]
Head and neck	Prospective	79	Correlation with xerostomia (*p* = 0.035)	[44]
*Lung*	*Prospective multicenter*	*561*	*Data pending*	[50,51]

**Table 2 cancers-14-02097-t002:** Available radiosensitivity assays with their respective level of evidence (based on the REMARK guidelines [69]). SNP: single nucleotide polymorphism, RILA: Radiation-Induced Lymphocyte Apoptosis.

Assay.	Tissue Sample	Level of Evidence	References
rs17599026 and rs7720298 SNPs for prostate cancer	Blood sample	I (meta-analysis)	[68]
RILA	Blood sample	I (prospective multicenter analysis)	[37,39,43,44,45,46,66,70,71]
SNPs for breast cancer	Blood sample	II (observational studies)	[72,73]
SNPs for lung cancer	Blood sample	II (observational studies)	[74,75]
Fibroblast-based assays	Skin biopsy	IV (retrospective studies)	[18,21,22,76,77,78]
G0 micronuclei	Blood sample	IV (retrospective studies)	[79,80,81]
G2 metaphase	Blood sample	IV (retrospective studies)	[79,82,83]
Residual γ-H2AX foci	Blood sample	IV (no validation studies)	[46,70,84,85,86,87,88,89]

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
