# Peer review of "Improving Patients’ Life Quality after Radiotherapy Treatment by Predicting Late Toxicities"

_cancers, 2022, doi:10.3390/cancers14092097_

Round 1

Reviewer 1 Report

Identifying those at risk of developing severe radiation-induced toxicity after radiation therapy is truly an important public health issue. Their RILA technique would allow for personalized treatment of patients.
In this form, the manuscript is very complete and documented and could be published in the journal Cancers.

Author Response

Thank you very much for your review

Reviewer 2 Report

In present review, authors claims to develop a test which could predict the off effects of radiotherapy before the commencement of treatment, in turn helping patients to improve the outcomes by personalization. I have several reservations, my comments are appended as below:

  1. Authors should first discuss on the toxicities in selected cancers on radiotherapy.
  2. While discussing the treatment efficacy, should specify the selected cancer type. For instance, line 71, 141.
  3. Line 99, 227- justify with reference.
  4. Along with lymphocytes, do studies focus on apoptosis in cells of innate immune system as neutrophils which are more abundant?

5.Table 1: indicate statistical inference (HR, P value)

  1. Do studies focus on obesity and radiotherapy toxicities?
  2. Line 181- ref 12- elaborate.
  3. RILA assay- the readouts are not clear, what cutoff do authors propose to consider ?
  4. Every section should end with a brief summary and future direction statement. There should be future direction statement after conclusion.
  5. Molecular rationale for the RILA assay: authors should incorporate a figure for better representation.
  6. RILA compared to other radiosensitivity assays: authors should compare with few parameters as sensitivity, specificity, ease, cost, and clinical evidences.

Reviewer 3 Report

The review titled "Improving patients’ life quality after radiotherapy treatment by 2 predicting late toxicities" focus on predicting late toxicities after radiotherapy using RILA assay. The authors claim that though the mechanistic basis is unclear of RILA assay, it is still a robust method to help deciding best available treatment in radiotherapy planning. The review is mainly focused on RILA assay, providing background, experimental details, advantages, and limitation of RILA assay. This review is of interest to a broad readership of Cancers journal.

Minor suggestions: Figure 1 clarity could be improved, it looks blurred and in place of PI, it could be mentioned as propidium iodide. 

Author Response

We changed the figure 1 in order to improve the quality and we change PI to Propidium Iodide as suggested. Thank you very much.

Round 2

Reviewer 2 Report

All my comments are answered.

This manuscript is a resubmission of an earlier submission. The following is a list of the peer review reports and author responses from that submission.

Round 1

Reviewer 1 Report

In this article, the authors aimed to establish a reproducible procedure for the RILA assay that can be implemented in any laboratory, without the need to irradiate cells from patient blood and replace this irradiation with a radiomimetic agent which is bleomycin. The idea is indeed very interesting because if before the radiotherapy we can identify the patients have a risk of severe toxicity induced by radiation this will allow a personalized treatment and improve their quality of life in the long term.

The article is well written, the results, conclusion and discussion are well argued, but one negative point is that the introduction is far too long. In its current form, the article is not publishable.

One has the impression that in this article there is a mini review. Maybe the authors could do 2 articles?

In this introduction, there are many historical explanations but they should be shorter and more concise, there is too much information.

Also for the RILA technique (part 1.1), it is described in the material and method section, there is no need to go into details in the introduction and the diagram should be put in the material and method section not in the introduction.

Make a summary of parts 1.1 to 1.4, this is also way too detailed for an article describing experiments.

Finally, the introduction lacks a description of bleomycin and why it was chosen as an alternative to irradiation for this test to be used in laboratories.

In part 3.2 the pink and blue trajectories are really different in figure 2 between irradiation and bleomycin. How can we be sure that 152 µg/ml is really the dose to use for all patients?

Reviewer 2 Report

Dear Authors, Thank you for submitting your article “Improving patients’ life quality after radiotherapy treatment by 2 predicting late toxicities” for consideration in Cancers.

While the findings of the authors are interesting, there are several major concerns with regard to the experimental rigourI have several suggestions for the author to improve the manuscript and make it more complete.

  1. Why has the author not taken into consideration the differential effect of bleomycin?
  2. What about dose limiting toxicity among different patients? How is the author going to compare the bleomycin toxicity to the effect of radiation therapy among different patients?
  3. Question of timeframe to see the cytotoxic drug development of bleomycin and comparing it with the effect of 8G does not seem to be fit to compare the effect in a holistic approach. For the majority of agents show bone marrow suppression after 2- 4 weeks of initiation therapy, so increase in the time frame would be better to include the late toxic effects of the radiotherapy and for comparative studies too.
  4. Have you checked the baselines for acute tolerance with different doses of bleomycin and radiotherapy doses for CD8+T cells?

Reviewer 3 Report

in this work, the authors attempt to develop a rapid test to predict possible adverse patient responses to radiotherapy.  Identification of these patients could be of major importance to clinicians since five to ten percent of patients who received radiotherapy will develop side effects. To this, they propose to test the radiation-induced apoptosis in lymphocytes using a RILA test.  Unfortunately, they test the effects o radiotherapy and bleomycin only on 10 hearty donors. This is an extremely small number to be able to draw conclusions on human samples, which have a high heterogeneity. Also, the authors do not test lymphocytes from cancer patients. For this reviewer, the data included in this paper is insufficient to support the validity of the test.  

The introduction is too long, probably it can be used for a reviewer paper.  

In the Simple Summary, the take-home message is not clearly expressed.